# Field pea leaf disease classification using a deep learning approach

**Dagne Walle Girmaw** [1]*, **Tsehay Wasihun Muluneh**[2]

**1** Department of Information Technology, Haramaya University, Haramaya, Ethiopia, **2** Department of Information Technology, University of Gondar, Gondar, Ethiopia

* dagnewalle143@gmail.com

**Data Availability Statement:** The dataset that supports the findings of this study has been uploaded as supplementary information. Access to this dataset can be requested from the corresponding author upon reasonable request.

## Abstract

Field peas are grown by smallholder farmers in Ethiopia for food, fodder, income, and soil fertility. However, leaf diseases such as ascochyta blight, powdery mildew, and leaf spots affect the quantity and quality of this crop as well as crop growth. Experts use visual observation to detect field pea disease. However, this approach is expensive, labor-intensive, and imprecise. Therefore, in this study, we presented a transfer learning approach for the automatic diagnosis of field pea leaf diseases. We classified three field pea leaf diseases: Ascochyta blight, leaf spot, and powdery mildew. A softmax classifier was used to classify the diseases. A total of 1600 images of both healthy and diseased leaves were used to train, validate, and test the pretrained models. According to the experimental results, DenseNet121 achieved 99.73% training accuracy, 99.16% validation accuracy, and 98.33% testing accuracy after 100 epochs. we expect that this research work will offer various benefits for farmers and farm experts. It reduced the cost and time needed for the detection and classification of field pea leaf disease. Thus, a fast, automated, less costly, and accurate detection method is necessary to overcome the detection problem.

## 1. Introduction

Field pea (*Pisum sativum* L) is one of the most significant crops used and produced by smallholder farmers in Ethiopia [1, 2]. This crop is the second most produced legume in Ethiopia in terms of volume after fava beans [1]. This crop is crucial to the farming community in the highlands of Bale, southeastern Ethiopia, and it provides revenue and acts as a rotational crop. However, this crop is affected by several diseases, such as Ascochyta blight (Ascochyta pisi), powdery and downy mildew (Erysiphe polygoni), and leaf spot. These diseases are causes of yield loss in production. Ascochyta blight and powdery mildew are the two most prevalent field pea diseases in mid-altitudes, and these diseases can reduce yields by 20–30% [1, 3]. Worldwide, Ascochyta blights (Ascochyta spp.) significantly reduce field pea yields and degrade seed quality. Ascochyta blight often results in a total loss of crop production in Dembi, Ethiopia, where the disease is highly prevalent. The Kulumsa Agricultural Research Centre is an ideal location for this disease because the days are frequently dry and hot [1, 4]. Field peas are grown around the world and have 86% loss due to powdery mildew disease, which reduces crop production. In Ethiopia, this disease is a main cause of yield losses of 21.09% [2–4].

**Funding:** The author(s) received no specific funding for this work.

**Competing interests:** NO authors have competing interests

Therefore, proper detection of field pea leaf diseases is crucial to improve the quality and quantity of crop production [5]. Currently, experts use visual observations to detect field pea disease. Nevertheless, this method has drawbacks and is expensive for large farms. Farmers might also lack the necessary resources or even the idea to consult experts; thus, expert consultation is costly and time-consuming. In large farms, the cost of visually observing and detecting leaf diseases is high, inaccurate, and difficult [6, 7]. Therefore, we suggest a deep learning approach to classify diseases in field pea leaves. It resolves these issues with traditional image processing techniques. A deep learning method uses multiple layers to process the data and extracts information from the image [5, 8]

The structure of the paper is as follows: related works are explored in section 2. Methods and materials are presented in section 3. The results and discussion are presented in Section 4. A conclusion is stated in Section 5.

## 2. Related works

Several studies have been performed on plant disease detection. The authors used different approaches, such as image processing, machine learning, and deep learning.

[9] presented convolutional neural networks (CNNs) to identify diseases in rice and potato plants. The authors identified diseases such as brown spot, blast, bacterial blight, and tungro. Images of potato leaves are divided into three categories: healthy, early blight, and late blight. The study used 1500 images of potato leaves and 5932 images of rice leaves. The suggested CNN model achieved 99.58% accuracy in classifying rice images and 97.66% accuracy in classifying potato leaves.

In [10] deep convolution networks for plant disease detection based on deep transfer learning were employed. The suggested approach is sufficiently light to significantly lower the processing expenses. The proposed method shows a considerable boost in efficiency with reduced complexity.

The PlantVillage (https://www.kaggle.com/datasets/mohitsingh1804/plantvillage) dataset was used for the experiment, and 18453 diseased leaves were divided into 3 categories based on species and 15 types of classification. The suggested method achieved a 99.28% accuracy rate on the dataset.

[11] suggested a model for image binarization using Otsu's global thresholding technique to eliminate the image's background noise. The suggested method is based on a fully connected CNN to identify the three rice diseases. The model was trained using 4,000 image samples of each damaged leaf and 4,000 image samples of healthy rice leaves. The model achieved a 99.7% accuracy rate on the dataset.

In [12] the YOLOv4 deep learning model with image processing was employed in a hybrid approach using the faster R-CNN, SSD, VGG16, and YOLOv4 deep learning models to automatically determine the severity of leaf spot disease on sugar beetroot and classify it. A total of 1040 images were used for training and testing the hybrid method and to determine their severity, and a classification accuracy rate of 96.47% was achieved. The suggested hybrid method produces better results than analyzing data using only these models.

In [13] a convolution neural network (CNN) approach was proposed to evaluate potato diseases (Healthy, Black Scurf, Common Scab, Black Leg, and Pink Rot). A database with 5,000 images of potatoes (https://www.kaggle.com/datasets/mohitsingh1804/plantvillage) was employed. The proposed model was compared with R-CNN, VGG, AlexNet, and GoogLeNet through transfer learning. The deep learning method's suggested accuracy is higher than that of previous works and achieved accuracy rates of 100% and 99%, respectively.

[14] employed a transfer learning approach using InceptionV3, ResNet50, VGG16, Mobile-Net, Xception, and DenseNet121 to identify plant leaf disease using the PlantVillage dataset. A total of 11,370 images of healthy and unhealthy tomato and potato leaves were included in the dataset. The method achieved an accuracy of 98.83% using the MobileNet architecture.

In [15]"Convnets" were employed to classify and identify plant diseases. The data are collected from the PlantViallge dataset, which includes plant classes such as potato, pepper, and tomato. The model achieved accuracy rates of 98.3%, 98.5%, and 95% in detecting tomato, pepper, and potato leaf diseases, respectively.

In [16] the proposed ResNet-9 model was used to identify blight disease in images of potato and tomato leaves. A total of 3,990 initial training data samples were used from the Plant Village Dataset. The model was evaluated on the 133 images of the test set. A 99.25% test accuracy, an overall precision of 99.67%, an overall recall of 99.33%, and an overall F1-score value of 99.33% were achieved.

[17] presented an autonomous method for detecting plant leaf damage and identifying disease. The suggested method used DenseNet to classify the disease based on an image of a plant leaf. The suggested DenseNet model yielded 100% classification accuracy. A deep learning-based semantic segmentation is used to identify leaf damage. A 97% accuracy rate was obtained using semantic segmentation. Apple, grape, potato, and strawberry plants were detected in the experimental analysis.

In [18] a deep learning method was proposed to identify different plant diseases. The proposed model implementation processes include acquiring datasets, training, testing, and classification to categorize leaves as healthy or diseased. The work identified potato leaf disease using the KNN and CNN methods. The developed method achieved an accuracy of ~ 90% using CNN-based classification.

[19] a deep learning-based approach to crop disease detection was proposed. The detection and classification of diseases is performed using a convolutional neural network-based method. Two convolutional and two pooling layers are employed within the model. According to the experimental findings, the suggested model achieved 98% training accuracy and 88.17% testing accuracy compared with pretrained models (InceptionV3, ResNet 152, and VGG19).

In [20] a lightweight convolutional neural network called VGG-ICNN was proposed for use in plant-leaf disease identification. VGG-ICNN had a significantly smaller number than most other high-performing deep learning models. PlantVillage and Embrapa provide 38 and 93 categories, respectively. The proposed work achieved 99.16% accuracy on the PlantVillage dataset.

In [21] the suggested system for identifying rice plant diseases used a computer vision-based methodology using deep learning, machine learning, and image processing techniques. The approach detects diseases in rice fields, such as sheath rot, brown leaf spot, rice blast, bacterial leaf blight, and false smut. The diseased region of the rice plant is recognized using image segmentation after image preprocessing. To identify and categorize distinct types of rice plant diseases, convolutional neural networks and a support vector machine classifier are employed, and the proposed deep learning-based approach achieved the greatest validation accuracy of 0.9145 using ReLU and softmax algorithms.

[22] proposed a transfer learning approach for the identification of maize leaf diseases. A dataset of 18,888 images of both healthy and diseased leaves was classified using pretrained VGG16, ResNet50, InceptionV3, and Xception models. The findings show that all trained models can classify maize leaf diseases with an accuracy of greater than 93%. Specifically, Xception, InceptionV3, and VGG16 all attained accuracies greater than 99%. Finally, we summarized all the related work as presented in Table 1.

**Table 1. Summary of the related works.**

| Authors | Title | Method | Accuracy | Observed Weakness |
|---|---|---|---|---|
| [9] | Plant disease diagnosis and image classification using deep learning | Convolutional Neural Networks (CNNs) | 99.5% and 97.66% | The CNN model requires extensive training to get an acceptable result |
| [10] | SK-MobileNet: A Lightweight Adaptive Network Based on Complex Deep<br><br>Transfer Learning for Plant Disease Recognition | Transfer Learning | 99.28% | Complex Model requires large GPU and longer training times. |
| [11] | A novel approach for rice plant disease classification with deep convolutional neural network | Deep Learning with Otsu's global thresholding technique | 99.7% | Otsu's global thresholding requires a significant amount of computational work. |
| [12] | A sugar beet leaf disease classification method based on image processing and deep learning | Hybrid model | 96.47% | Hybrid models have an overfitting problem. |
| [13] | Potato disease detection and classification using deep learning methods | Convolution neural network (CNN) | 100% and 99% | The CNN model requires extensive training to get an acceptable result |
| [14] | Performance Analysis of Deep Learning Algorithms Toward Disease Detection: Tomato and Potato Plant as Use-Cases | Transfer learning | 98.83% | Inception V3 model is not appropriate for diseases exhibiting numerous lesions |
| [15] | Plant Disease Prediction and classification using Deep Learning ConvNets, | Deep Learning | 98.3%, 98.5%, and 95% | Such extensive architectures may result in poor convergence and overfitting. |
| [16] | Automatic blight disease detection in potato and tomato plants using deep learning | Transfer Learning (ResNet-9) | 99.25% | The model requires a large GPU and longer training times. |
| [17] | Plant leaf disease classification and damage detection system using deep learning models | Transfer Learning (DenseNet) | 97% | The model can cause significant features to be skipped or lost. |
| [18] | Deep Learning-Based Approach to Identify the Potato Leaf Disease and Help in Mitigation Using IOT | Convolutional Neural Networks and KNN | 90% | The CNN model requires extensive training to get an acceptable result |
| [19] | Detection and Classification of Tomato Crop Disease Using Convolutional Neural Network | Convolutional Neural Network | 88.17% | The CNN model requires extensive training to get an acceptable result |
| [20] | VGG-ICNN: A Lightweight CNN model for crop disease identification | A Lightweight CNN model | 99.16% | The model requires a large GPU and longer training times. |
| [21] | Deep learning system for paddy plant disease detection and classification | Deep learning and SVM | 91.4% | For noisy image data, the support vector machine approach is inappropriate. |
| [22] | On fine-tuning deep learning models using transfer learning and hyperparameters optimization for disease identification in maize leaves | Transfer learning | 99% | When the FTNN technique is fed new weights, they forget the old weight that was connected with them, which could affect the result |

# 3. Methods and materials

These are the descriptions of methods and materials undertaken in the study:

**Image pre-processing:** To resize and rescale the input images, image preprocessing is necessary. Before feeding the image into the model, it should be appropriately sized. We resized and rescaled the image using the image data generator method. We used data augmentation techniques such as rotation, horizontal flip, zoom, and shear to diversify training images [23] and enhance the model's performance [24]. Images are resized to a common size of 224x224 pixels and normalized in the range [0, 1] to scale pixel values [25, 26].

**Model Training:** We found the optimal hyperparameters such as optimizer, learning rate, and batch size through experimentation. We used dropout and batch normalization to prevent overfitting problems. To train the model we used an optimizer, loss function, number of epochs, and early stopping to stop the training when performance on the validation set starts failing.

**Evaluation:** We used evaluation metrics such as accuracy, precision, recall, F1 score, assess model performance.

**Deployment:** We saved trained models and deployed them as web applications using the Flask application. This allows the user to perform a live field pea leaf disease classification service.

The suggested work contains three phases: training, validation, and testing.

### 3.1 Training phase

To train the pretrained model, we labeled the images of field peas with the appropriate class, and the model was trained using labeled images. 1600 field pea images were labeled for this study. Field pea images were obtained from the Kulumsa Agricultural Research Centre in Ethiopia. A smartphone camera was utilized to take images. The study was conducted at Kulumsa Agricultural Research Centre in Ethiopia.

### 3.2 Validation phase

Before the input images feed into the model, we employed image pre-processing techniques such as image resizing, normalization, and argumentations. To validate the pretrained models, we employed of these approaches.

### 3.3 Testing phase

The models were tested using the test images to evaluate the model performance.240 images of field peas are used for testing. Test images were obtained from the same dataset used for training and validation

### 3.4 Transfer learning

We used transfer learning to retrain a previously trained model for a new problem. Transfer learning offers numerous advantages, including the ability to finetune parameters quickly and easily to learn new tasks without defining a new network. We adopted a transfer learning technique for the classification of pea leaf diseases as follows:

**Choosing a Pre-trained Model**: We selected novel pre-trained models (EfficentNetB7, MobileNetV2, and DenseNet121) for the classification of pea leaf diseases.

Loading the Pre-trained Model: We loaded the selected pre-trained model weights and excluded the top classification layers because they are specific to the original task for which the model was trained.

**Adding Custom Classification Layers:** We added custom layers for our classification task of pea leaf disease and these layers are followed by a SoftMax classifier.

**Freezing Pre-trained Layers:** We freeze these layers to stop the weights of the pre-trained layers from being updated during training and to train only the weights of the newly added layers for our classification task.

**Compiling the Model:** The model was compiled using Adam optimizer and categorical cross-entropy as a loss function.

**Data Augmentation:** Data augmentation techniques such as rotation, horizontal flip, zoom, and shear are employed to diversify training images and enhance the model's performance.

**Training the Model:** The model was trained using training data and the weights of the recently added classification layers are updated by the backpropagation method.

**Evaluation:** After training is finished, the model's performance is assessed using a test dataset and we use accuracy, precision, recall, and F1-score to evaluate the model's performance.

In this study, we employed a transfer learning approach to classify field pea disease, which enabled us to quickly and readily adjust parameters to learn new tasks without creating a new network [27–29]. It trained to classify images into 1000 classes using more than a million images [30]. For our classification task, we reuse the pretrained models (EfficentNetB7,

MobileNetV2, and DenseNet201). All layers are maintained, and we use the final layer of the model for our detection task.

**3.4.1 MobileNet.**   The goal of the MobileNet model is to improve deep learning's real-time performance with limited resources. It uses fewer computational resources than the other CNN models, and it is perfect for usage on computers with low processing power as well as mobile devices [31]. A 1x1 convolution and a type of factorized convolution or depthwise separable convolution serve as the foundation for the model architecture. Every input channel receives a single filter application from the depthwise convolution. The outputs of the depthwise convolution are combined with the pointwise convolution using 1x1 convolutions [10].

**3.4.2 DenseNet.**   DenseNet works using a densely layered design with direct connections. In comparison to traditional convolutional neural networks, it uses fewer parameters [32, 33]. It is employed in CNN networks to streamline the layer-to-layer connectivity structure. It fixes the problems caused by the gradients, and easy access to the input images is provided for each layer of the model [34].

**3.4.3 EfficientNet.**   The EfcientNet model stretches the network using an inverted bottleneck convolution and a swish activation function. It reduces the calculation time by the square of a filter size [30, 35]. It has a better classification performance than other deep CNN models in terms of accuracy [36]. The proposed work is presented in Fig 1.

## 3.5 Performance metrics

We assessed the model performance using ROC curve, support, precision, recall, and accuracy. The following is a mathematical expression of the metrics as presented in Table 2.

## 4. Results and discussion

### 4.1 Dataset Acquisition

We collected 1600 images from the Kulumsa Agricultural Research Centre with the help of agricultural experts. All images were resized to 224 x 224, and we used 15% of the data for

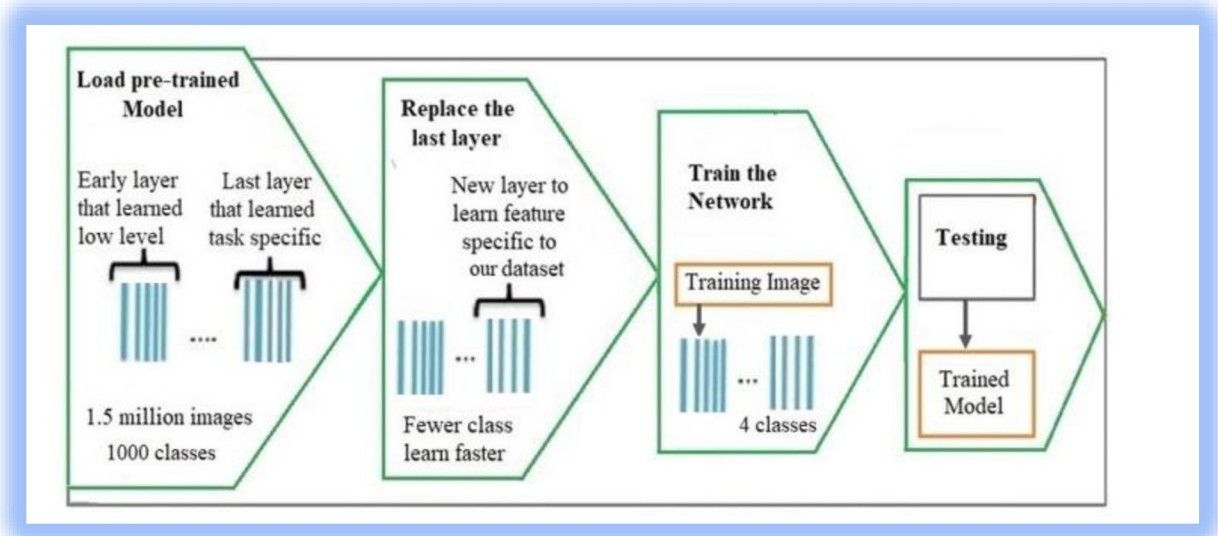

**Fig 1.  The proposed methodology.**

**Table 2. Performance metrics.**

| Metric | Definition | Symbol | Reference |
|---|---|---|---|
| | | A | [37] |
| Accuracy | $A = \frac{TP+TN}{TP+FN+TN+FP}$ | | |
| | Where | | |
| | TP = true positive | | |
| | TN = true negative | | |
| | FP = false positive | | |
| | FN = false negative | | |
| Precision | $P = \frac{TP}{TP+FP}$ | P | |
| Recall | $R = \frac{TP}{TP+FN}.$ | R | |
| F1-score | F1-Score $= \frac{2 \times P \times R}{P+R}.$ | F1 | |

testing,15% of the data for validating, and 70% of the data for training. The dataset is divided into four classes that correspond to the diseases. The distribution of classes is as follows: the first class consists of 400 images of ascochyta blight disease, the second class consists of 400 Healthy images, the third class consists of 400 images of leaf spot disease and the fourth class consists of 400 images of powdery mildew leaves. Images are resized to a common size of 224x224 pixels and normalized in the range [0, 1] to scale pixel values. The summary of the dataset is presented in Table 3.

## 4.2 Environment setup

Models were trained and tested on Google Colab, which provided free resources such as GPU and RAM. We wrote the code using Jupyter Notebook on a computer with a 64-bit operating system, Intel(R) Core i3 processor, and 4 GB of RAM. we used GPUs to accelerate training time, TensorFlow, and Keras, for building and training deep learning models, Jupyter Notebook for writing, debugging, and running deep learning code, Matplotlib, Seaborn, and Tensor Board for visualizing training metrics, model architectures, and data distributions.

## 4.3 Pretrained model training

Initially, we downloaded the MobileNetV2 model from 'Keras. We retrained the last layer of the model for our disease classification, and we used batch sizes of 32 for 100 epochs. The model achieved 99.64% training accuracy, 98.33% validation accuracy, and a testing accuracy of 96. 09%. The training, validation, and loss accuracies are displayed in Figs 2 and 3.

In addition, we downloaded the DenseNet 121 model from 'keras. We retrained the last layer of the model with 32 batch sizes and 100 epochs. The model achieved a 99.73% training accuracy, a 99.16% validation accuracy, and a testing accuracy of 98.33% after 100 epochs. Figs 4 and 5 display its training accuracy, validation accuracy, and losses.

**Table 3. Summary of the dataset.**

| No | Class | Total number of images | Source of images |
|---|---|---|---|
| 1 | Ascochyta blight | 400 | Kulumsa Agricultural Research Centre |
| 2 | Powdery mildew | 400 | Kulumsa Agricultural Research Centre |
| 3 | Leaf spots | 400 | Kulumsa Agricultural Research Centre |
| 4 | Healthy | 400 | Kulumsa Agricultural Research Centre |

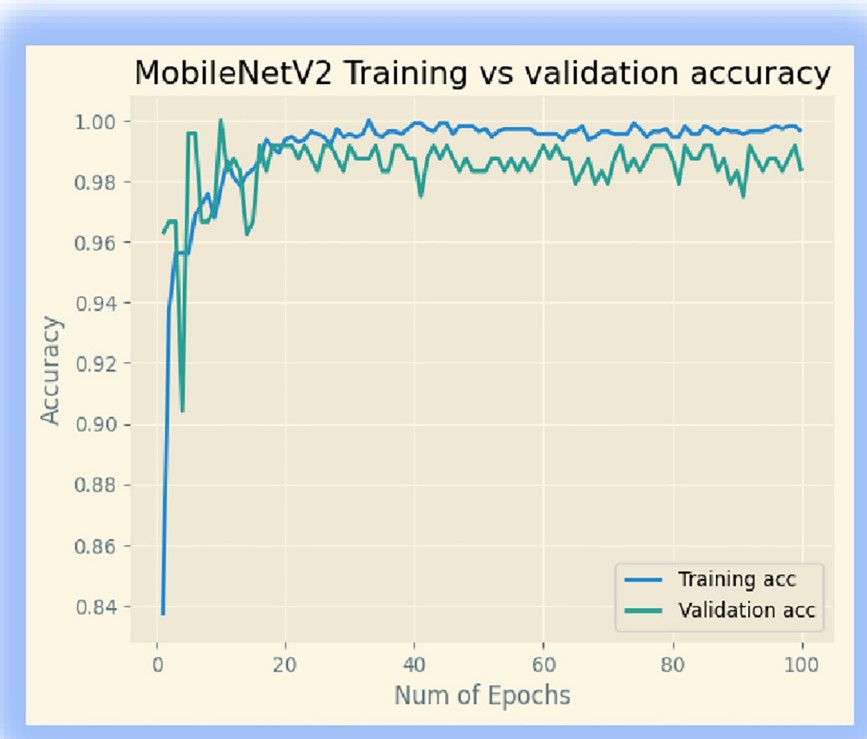

**Fig 2. Training and validation accuracy.**

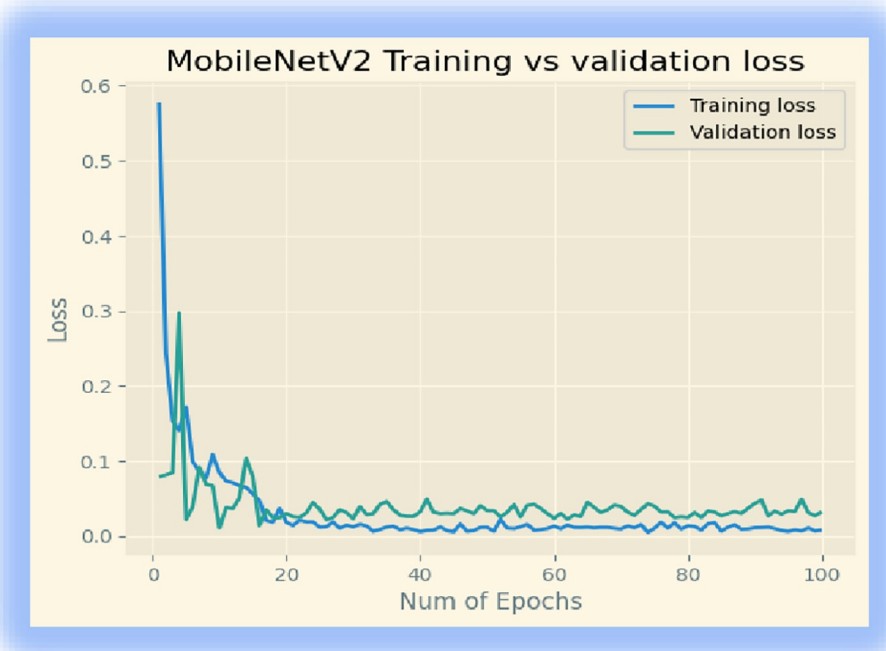

**Fig 3. Training and validation loss of the model.**

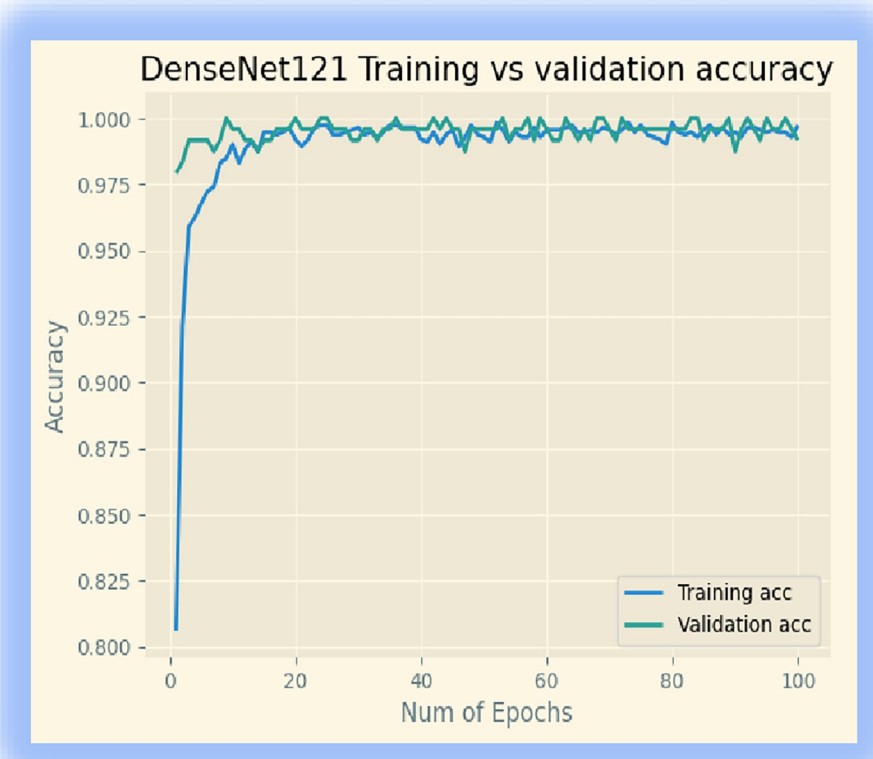

**Fig 4. Training accuracy and validation accuracy.**

In the third experiment, we downloaded the EfficientNetB7 model from 'Keras. We retrained the last layer of the model using 32 batch sizes and 100 epochs. The model achieved a training accuracy of 99.82%, a validation accuracy of 99.16%, and a testing accuracy of 97.92%. Figs 6 and 7 display its training accuracy, validation accuracy, and losses.

We made model comparisons in terms of accuracy, loss, and receiver operating characteristic (ROC) curves for the classification of field pea leaf diseases, as presented in Figs 8 and 9.

### 4.4 Visualizing of channels during the activation layer

Channel visualization provides an overview of how deep learning divides any input into discrete filters. Two arguments are used to instantiate a model: input and output tensor. To visualize it, an input image is fed to the model, and it returns the layer's activation values. The first layer of the network in the visualization has multiple detectors, including edge, bright dot, and brightness detectors. The feature maps in this layer hold all of the information related to the input image. Nonimportant characteristics were skipped in the first few layers. When we go deeper and deeper inside the neural network, we learn more abstract features and cannot interpret what the filter is doing, as presented in Fig 10.

### 4.5 Model deployment using a flask

A flask application was used to launch the suggested model as a web application. This enables a real-time field pea leaf disease classification service to the user, as presented in Figs 11–13.

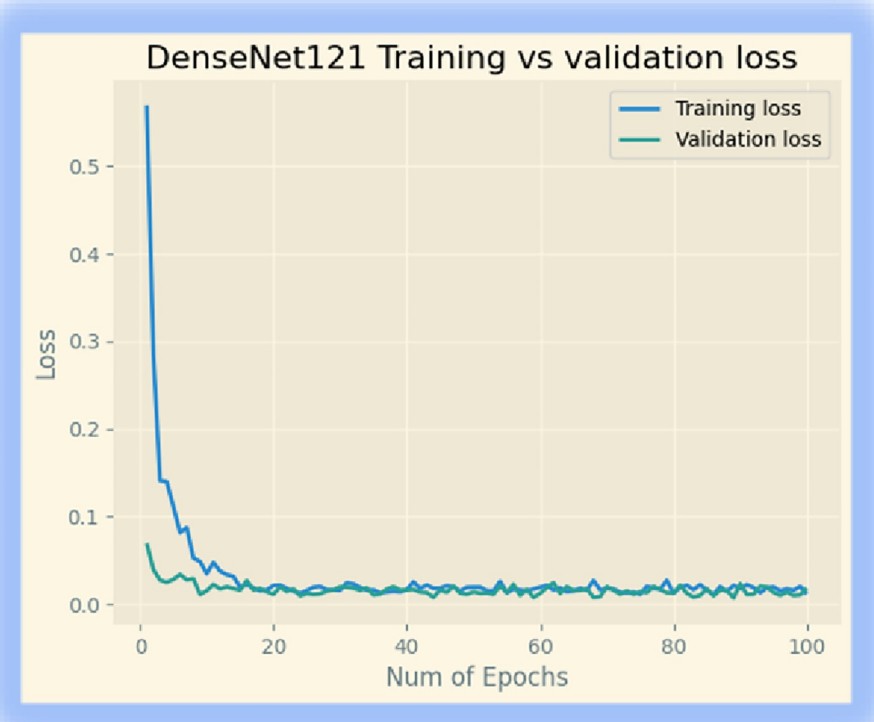

**Fig 5. Training and validation losses.**

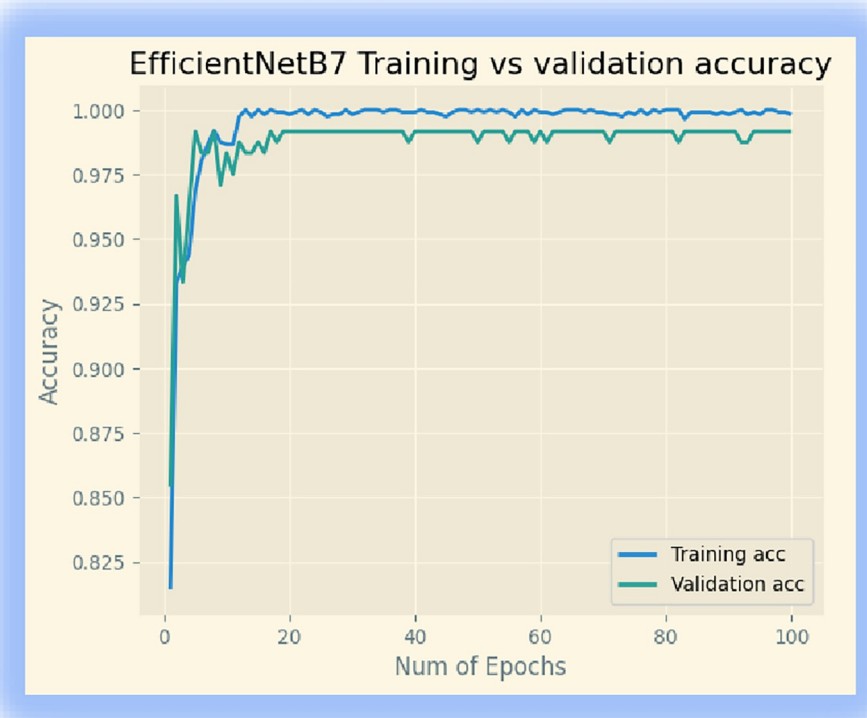

**Fig 6. Training accuracy and validation accuracy.**

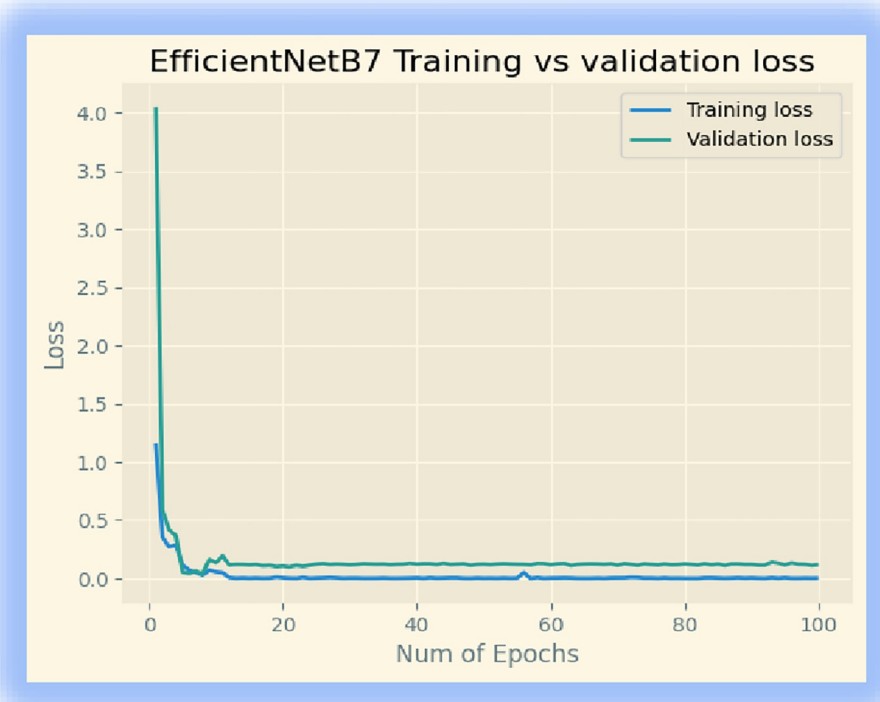

**Fig 7. Training and validation losses.**

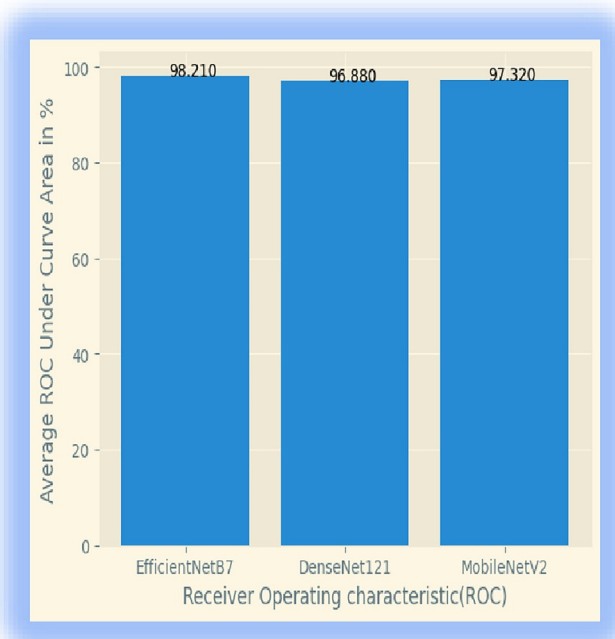

**Fig 8. ROC curves of the three models.**

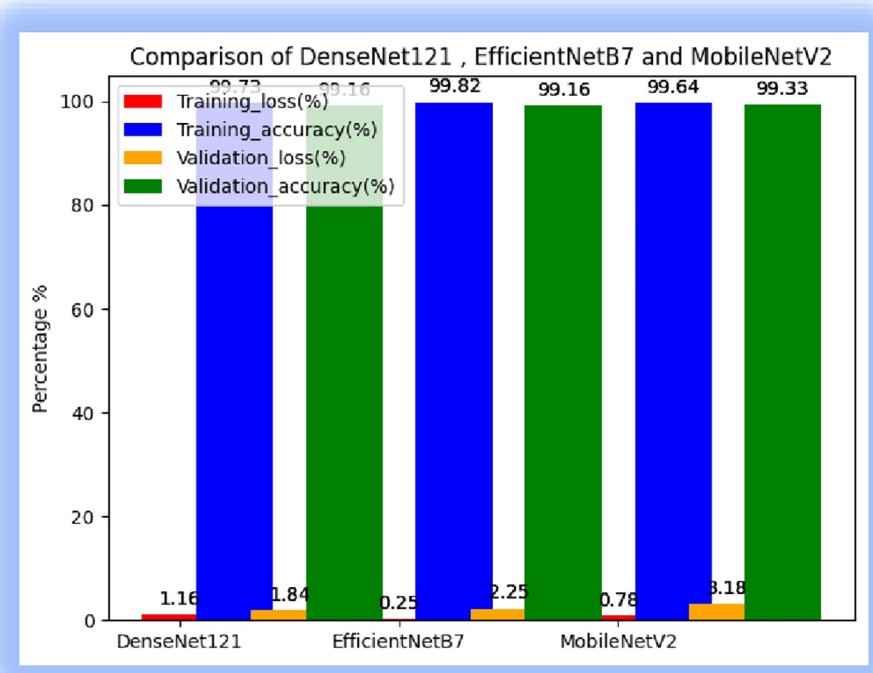

**Fig 9. Model comparison.**

## 4.6 Discussion

From the state-of-the-art models, DenseNet121 achieved a training accuracy of 99.73% validation accuracy of 99.16%, and testing of 98.33%% and MobileNetV2 scored a training accuracy of 99.64%, validation accuracy of 98.33%and testing of 96. 09%, While EfficientNetB7 scored a training accuracy of 99.82%, validation accuracy of 99.16%, and testing of 97.92%. The experimental results showed that DenseNet121 performed better than MobileNetV2 and EfficientNetB7 for field pea leaf disease detection. The experimental results also validated the use of

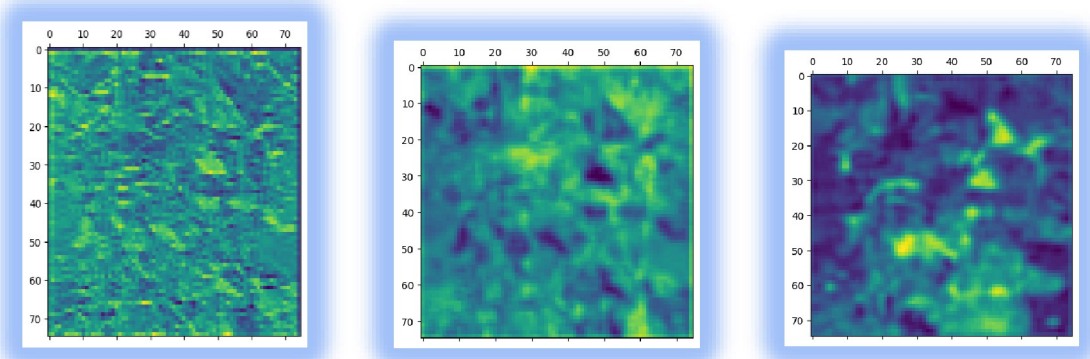

(A) Visualization in the 1$^{st}$ channel (B) Visualization in the 2$^{nd}$ channel (C) Visualization of activation in the 3$^{rd}$ channel

**Fig 10. Visualizations of the activation of channels.** (A) Visualization in the 1$^{st}$ channel (B) Visualization in the 2$^{nd}$ channel (C) Visualization of activation in the 3$^{rd}$ channel.

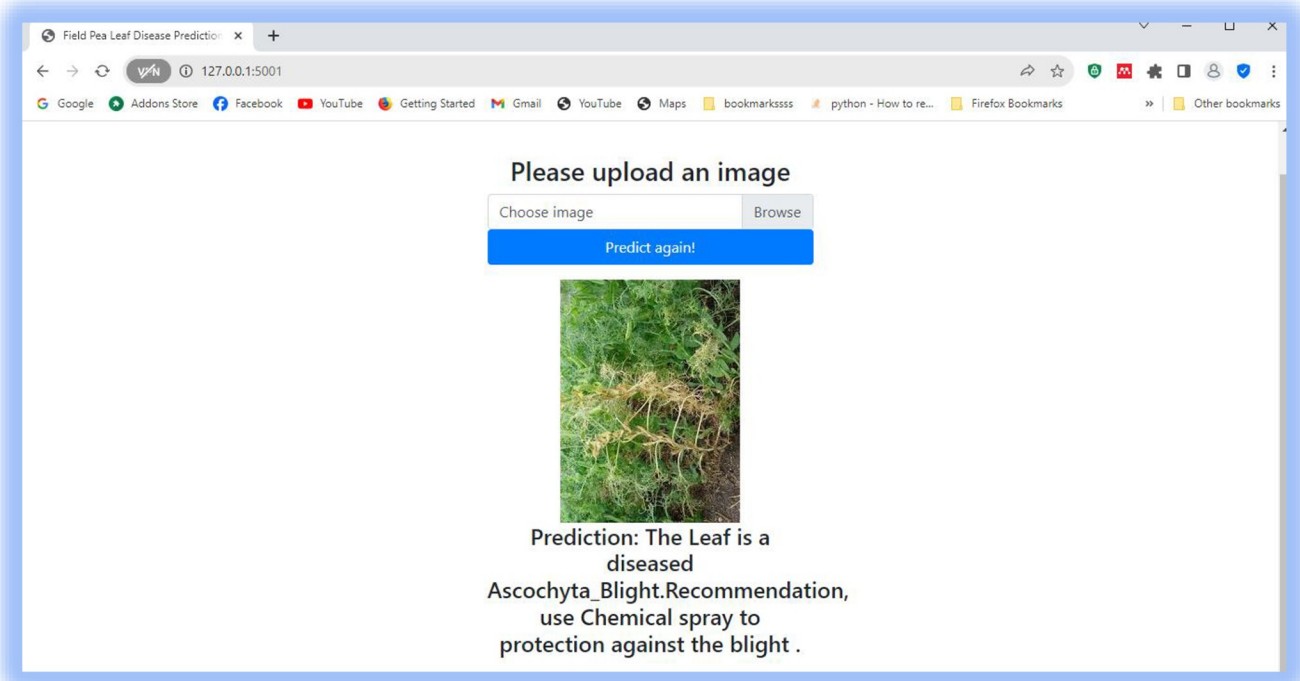

**Fig 11. Predicting field pea blight.**

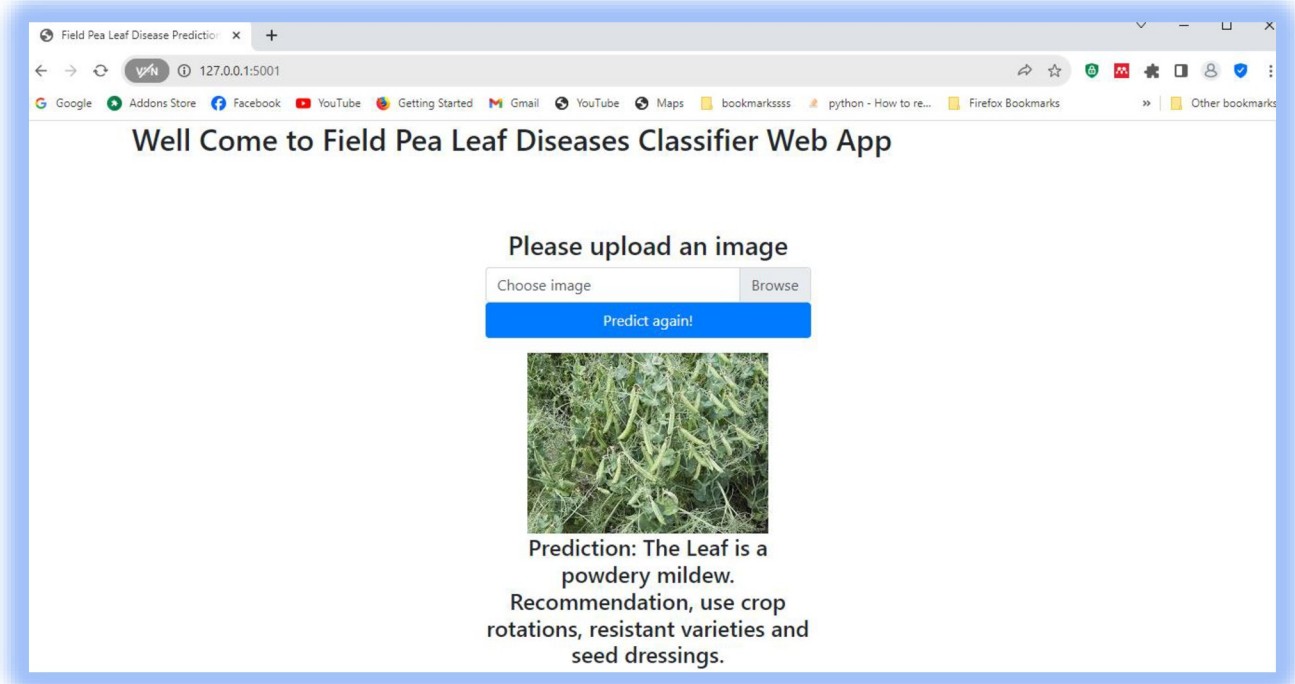

**Fig 12. Predicting field pea powdery mildew.**

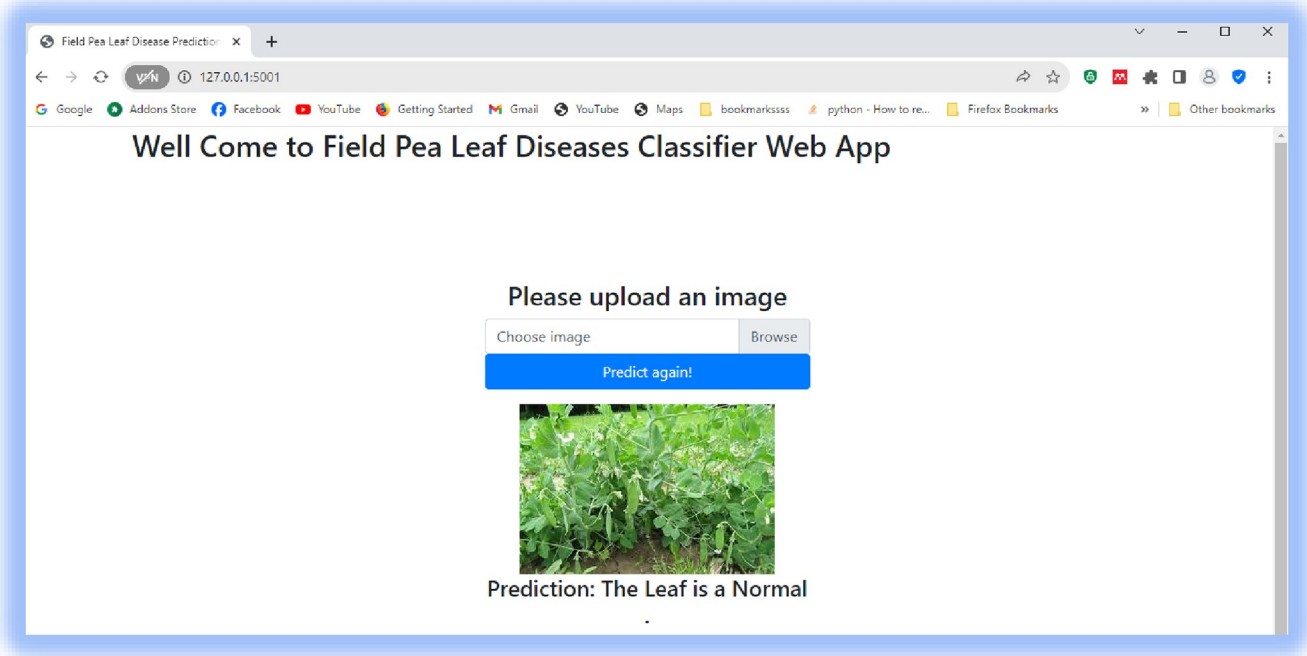

**Fig 13. Predicting field pea normal.**

transfer learning for field pea leaf disease classification. Overfitting was one challenge to adopting transfer learning for leaf disease classification when the target dataset is small. This means the model may memorize the training data rather than learning features. we mitigated this issue using data augmentation and dropout techniques. fine-tuning the hyperparameters of the pre-trained model was another challenge to determine the optimal parameters such as learning rate, optimizer, and dropout value

## 5 Conclusion

Hence, we proposed a work using transfer learning for field pea disease classification. The main contributions of this study are as follows: Performance improvement: We improved the performance of the proposed work using recent deep-learning models. Experimental results show that DenseNet121 achieves 98.33% testing accuracy. Data collection: We collected a dataset of field pea lea diseases from Kulumsa Agricultural Research Centre with the help of farm area experts. Medical treatment recommendation: We deployed the model using a flask application to detect the disease and suggest a medical treatment. Visualization: It enables new users to know how deep learning models work internally to classify the disease. Future works are: Exploring the performance of other pre-trained models such as AlexNet, GoogleNet, and VGG-19 for filed pea leaf diseases classification task. Incorporating the textual features of symptoms on field pea leaves to enhance classification accuracy. Applying domain-specific data augmentation techniques to replicate changes in disease severity and leaf orientation.

## Supporting information

**S1 Dataset.**
(ZIP)

## Author Contributions

**Data curation:** Dagne Walle Girmaw, Tsehay Wasihun Muluneh.

**Methodology:** Dagne Walle Girmaw, Tsehay Wasihun Muluneh.

**Software:** Dagne Walle Girmaw.

**Validation:** Dagne Walle Girmaw, Tsehay Wasihun Muluneh.

**Visualization:** Dagne Walle Girmaw.

**Writing – original draft:** Dagne Walle Girmaw.

**Writing – review & editing:** Tsehay Wasihun Muluneh.

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
