## [Decision Letter · Decision Letter 0]

9 Jun 2024

PONE-D-24-10799Field Pea Leaf Disease Classification Using a Deep Learning ApproachPLOS ONE

Dear Dr. Girmaw,

Thank you for submitting your manuscript to PLOS ONE. After careful consideration, we feel that it has merit but does not fully meet PLOS ONE’s publication criteria as it currently stands. Therefore, we invite you to submit a revised version of the manuscript that addresses the points raised during the review process.

We look forward to receiving your revised manuscript.

Kind regards,

Valentine Otang Ntui, Ph.D

Academic Editor

PLOS ONE

3. In the online submission form, you indicated that [Insert text from online submission form here].

5. Please include your tables as part of your main manuscript and remove the individual files. Please note that supplementary tables (should remain/ be uploaded) as separate "supporting information" files.

Reviewers' comments:

Reviewer's Responses to Questions

**Comments to the Author**

1. Is the manuscript technically sound, and do the data support the conclusions?

Reviewer #1: Yes

Reviewer #2: Yes

2. Has the statistical analysis been performed appropriately and rigorously? 

Reviewer #1: N/A

Reviewer #2: Yes

3. Have the authors made all data underlying the findings in their manuscript fully available?

Reviewer #1: Yes

Reviewer #2: Yes

4. Is the manuscript presented in an intelligible fashion and written in standard English?

Reviewer #1: No

Reviewer #2: Yes

5. Review Comments to the Author

Reviewer #1: The manuscript by the author addresses detection of three pea plant diseases and classification using deep learning approach. In my view, I recommended this article with some revisions.

The author can revise the manuscript including details on the following:

1. The images quality can be improved. Some sections of the paper could be revised for clarity and conciseness. Additionally, attention should be paid to grammar, spelling, and formatting of text to enhance the overall readability of the paper.

2. Detailed explanation of the methodology adopted for transfer learning can be included. It would be beneficial to provide a step-by-step explanation of how the pre-trained model was adapted and fine-tuned for pea leaf disease classification.

3. More details are needed regarding the dataset used for training, validation, and testing. It would be helpful to include information on sample images of diseased pea leaf, distribution of classes, and any preprocessing techniques applied.

4. The paper would be strengthened by including comparisons with other classification approaches or baselines. This could demonstrate the effectiveness of the transfer learning approach in comparison to traditional methods or other deep learning architectures.

5. The results and discussion should include interpretation and discussion of the findings. It would be beneficial to analyze the performance of the model in classifying different types of pea leaf diseases and discuss any challenges or limitations encountered.

6. Suggestions for future work could be provided to guide further research in this area. This could include exploring different pre-trained models, incorporating additional features or data augmentation techniques.

Reviewer #2: Generally, the manuscript is well-written and provides a novel method for diagnosing pea leaf diseases. The authors have provided all the data underlying the findings of the study. The paper is also presented in an intelligible fashion using standard English. However, the materials and methods section has only provided limited information on the protocols undertaken in the study.

6. PLOS authors have the option to publish the peer review history of their article (what does this mean?). If published, this will include your full peer review and any attached files.

Reviewer #1: No

Reviewer #2: **Yes: **Duncan Njora Waweru

---

## [Author Response · Author response to Decision Letter 0]

17 Jun 2024

REVIEWED PAPER TITLE:

Ms. Ref. No.: PONE-D-24-10799

Paper Title: Field Pea Leaf Disease Classification Using a Deep Learning Approach

Journal: PLOS ONE.

Responses to Reviewer’s/Editor’s Comments

Acknowledgment from the Authors: We are sincerely grateful to the 

reviewers and Editorial Board for the constructive comments tailored to improve the

quality of the manuscript. We are also thankful for the comments/suggestions for further improvement/reconsideration of our manuscript. We deeply acknowledge the efforts

made by the reviewers for devoting valuable time to improving our

manuscript. The changes made concerning the reviewer’s suggestions are highlighted in color red for Reviewers in the revised manuscript with track changes comments from the Editors and Reviewers: 

Reviewer’s comments:

 1. Is the manuscript technically sound, and does the data support the conclusions? The manuscript must describe a technically sound piece of scientific research with data that supports the conclusions. Experiments must have been conducted rigorously, with appropriate controls, replication, and sample sizes. The conclusions must be drawn appropriately based on the data presented

Response from authors: Yes, we provide a novel method for diagnosing pea leaf disease, and achieve good performance on the dataset. The research follows design science research with five basic steps such as problem identification, design solution, development, demonstration, evaluation, and communication. 

2. Has the statistical analysis been performed appropriately and rigorously? 

Response from authors: Yes, the study design is appropriate for the research question and objectives. The results of the statistical analysis are interpreted by using performance evaluation metrics such as accuracy, precision, recall, support, macro average, and F1 score.

3. Have the authors made all data underlying the findings in their manuscript fully available? 

 Response from authors: Yes.

4. Is the manuscript presented in an intelligible fashion and written in standard English? 

Response from authors: Yes, we corrected some irregular format, inconsistent statements, absence of content, and typos, in the entire section of the manuscript. Right now, the manuscript is written in clear, and brief language.

Review Comments to the Author: 

Reviewer #1. The manuscript by the author addresses the detection of three pea plant diseases and classification using a deep learning approach. In my view, I recommend this article with some revision. The author can revise the manuscript including details on the following:

1. The image quality can be improved. Some sections of the paper could be revised for clarity and conciseness. Additionally, attention should be paid to grammar, spelling, and formatting of text to enhance the overall readability of the paper.

Response from authors: Yes, the quality of images is improved to a higher resolution. The manuscript is written in clear, brief language to enhance the overall readability of the paper.

2. A detailed explanation of the methodology adopted for transfer learning can be included. It would be beneficial to provide a step-by-step explanation of how the pre-trained model was adapted and fine-tuned for pea leaf disease classification.

Response from authors: Yes, we adopted a transfer learning technique for the classification of pea leaf diseases as follows: 

Choosing a Pre-trained Model: We selected novel pre-trained models (EfficentNetB7, MobileNetV2, and DenseNet121) for the classification of pea leaf diseases. 

Loading the Pre-trained Model: We loaded the selected pre-trained model weights and excluded the top classification layers because they are specific to the original task for which the model was trained.

Adding Custom Classification Layers: We added custom layers for our classification task of pea leaf disease and these layers are followed by a SoftMax classifier.

Freezing Pre-trained Layers: We freeze these layers to stop the weights of the pre-trained layers from being updated during training and to train only the weights of the newly added layers for our classification task. 

Compiling the Model: We compiled the model using Adam optimizer and categorical cross-entropy as a loss function. 

Data Augmentation: We used data augmentation techniques such as rotation, horizontal flip, zoom, and shear to diversify training images and enhance the model's performance.

Training the Model: We trained the model using training data and the weights of the recently added classification layers are updated by the backpropagation method.

Evaluation: After training is finished, we assess the model's performance using a test dataset and we use accuracy, precision, recall, and F1-score to evaluate the model's performance. 

3. More details are needed regarding the dataset used for training, validation, and testing. It would be helpful to include information on sample images of diseased pea leaves, distribution of classes, and any pre-processing techniques applied.

Response from authors: The dataset is divided into four classes that correspond to the diseases. The distribution of classes is as follows: the first class consists of 400 images of ascochyta blight disease. The second class consists of 400 Healthy images. The third class consists of 400 images of leaf spot disease. The fourth class consists of 400 images of powdery mildew leaves. Images are resized to a common size of 224x224 pixels and normalized in the range [0, 1] to scale pixel values. We applied data augmentation to improve model generalization and increase the diversity of training data.

4. The paper would be strengthened by including comparisons with other classification approaches or baselines. This could demonstrate the effectiveness of the transfer learning approach in comparison to traditional methods or other deep learning architectures.

Response from authors: we incorporated a comprehensive comparison under related works to evaluate our approach against traditional methods and other deep learning architectures. 

5. The results and discussion should include interpretation and discussion of the findings. It would be beneficial to analyze the performance of the model in classifying different types of pea leaf diseases and discuss any challenges or limitations encountered.

Response from authors: Overfitting was one challenge to adopting transfer learning for leaf disease classification when the target dataset is small. This means the model may memorize the training data rather than learning features. we mitigated this issue using data augmentation and dropout techniques. fine-tuning the hyperparameters of the pre-trained model was another challenge to determine the optimal parameters such as learning rate, optimizer, and dropout value 

6. Suggestions for future work could be provided to guide further research in this area. This could include exploring different pre-trained models and incorporating additional features or data augmentation techniques.

Response from authors: Yes, here are some recommendations: 

Exploring Different Pre-Trained Models: Exploring the performance of other pre-trained models such as AlexNet, GoogleNet, and VGG-19 for filed pea leaf diseases classification task.

Incorporating Additional Features: Incorporating the textual features of symptoms on field pea leaves to enhance classification accuracy.

Data Augmentation Techniques: Applying domain-specific augmentation techniques to replicate changes in disease severity and leaf orientation.

Reviewer #2: Generally, the manuscript is well-written and provides a novel method for diagnosing pea leaf diseases. The authors have provided all the data underlying the findings of the study. The paper is also presented in an intelligible fashion using standard English. However, the materials and methods section has only provided limited information on the protocols undertaken in the study: 

Response from authors: Yes, these are the descriptions of protocols undertaken in the study

Data Pre-processing Protocol: We used data augmentation techniques such as rotation, horizontal flip, zoom, and shear to diversify training images and enhance the model's performance. Images are resized to a common size of 224x224 pixels and normalized in the range [0, 1] to scale pixel values. The dataset is divided into training, validation, and test sets to assess model performance.

Model Training Protocol: We found the optimal hyperparameters such as optimizer, learning rate, and batch size through experimentation. We used dropout and batch normalization to prevent overfitting problems. To train the model we used optimizer, loss function, number of epochs, and early stopping to stop the training when performance on the validation set starts failing.

Evaluation Protocol: We used evaluation metrics such as accuracy, precision, recall, F1 score, assess model performance. 

Deployment Protocol: We saved trained models and deployed them as web applications using the Flask application. This allows the user to perform a live field pea leaf disease classification service.

Comments

Abstract:

You have not indicated the significance of your study, which is usually the last statement in the abstract section.

Response from authors: Yes, we expect that this research work will offer various benefits for farmers and farm experts. It reduced the cost and time needed for the detection and classification of field pea leaf disease. Thus, a fast, automated, less costly, and accurate detection method is necessary to overcome the detection problem. 

Introduction:

Line 35: ‘Field peas are grown around the world….’ This has already been indicated, so there is no need to repeat it. The next statement ‘…and have 86% loss due to powdery mildew disease’ is contradictory to what you alluded to earlier that ‘…...these diseases can reduce yields by 20–30%’ (line 31). I suggest that you amalgamate both sentences and have a unified statement offering the specific statistics with a relevant citation.

Response from authors: Field peas, grown globally, are highly vulnerable to powdery mildew disease and cause substantial losses [3], [5]. Although it's commonly estimated that diseases like powdery mildew result in yield decreases of 20–30%, the actual losses can be even higher [1], [3].

Related works:

Line 63: ‘The Plant-Village dataset was used…’ Provide a link to indicate the dataset used. The same should be applied to line 78: ‘A database with 5,000 images of potatoes was employed.’ Line 93: ‘…on the 1,33 images of the test set.’ Is it 133 images? I don’t understand why you indicated the contributions of your work in the related works section before providing the methods and results obtained. This would have been appropriate for the conclusions of your study.

Response from authors: 

Line 63: The Plant-Village dataset (https://www.kaggle.com/datasets/mohitsingh1804/plantvillage) was used.

Line 78: A database with 5,000 images of potatoes (https://www.kaggle.com/datasets/mohitsingh1804/plantvillage) was employed.

Line 93: on the 133 images of the test set.

Contributions: we moved the discussion of contributions to the conclusions section for better organization.

Methods and materials:

Shouldn’t it be materials and methods? You need to indicate the materials used in the study. Generally, this section has provided limited information on the protocol you used. For example, in line 148: ‘We labeled the images of field peas…’ How many images? How were they obtained? Which camera was used to capture the images? Where was the study conducted? Line 151: ‘We employed several approaches, such as…’ Please specify the precise approaches used in the study. Line 154: ‘Models were tested using the test images…’ How many images? Where were they obtained? Did you use a specific field pea cultivar?

Response from authors: 

Labelling of Field Pea Images (Line 148):

Number of Images: 1600 field pea images were labeled for this study.

Acquisition: Field pea images were obtained from the Kulumsa Agricultural Research Centre in Ethiopia. 

Camera: A smartphone camera was utilized to take images. 

Study Location: The study was conducted at Kulumsa Agricultural Research Centre in Ethiopia.

Description of Approaches Used (Line 151): 

Before the input images feed into the model, we employed image pre-processing techniques such as image resizing, normalization, and argumentations. 

Testing Data Information (Line 154): 

Number of Test Images: 240 images of field peas are used for testing. 

Source of Test Images: Test images were obtained from the same dataset used for training and validation.

Results and discussion:

The information lacking in the methods section is provided in the results section, which is not appropriate. This section should only provide the results you obtained from the study and not how the study was conducted. I have also not seen any discussion of your results in comparison to previous studies. Generally, you have provided limited/incomplete information in the methods, results, and discussion sections.

Response from authors: 

Methods Section: This research is a design science, so we have built artifacts to detect and classify field pea leaf diseases. we used GPUs to accelerate training time, TensorFlow, and Keras, for building and training deep learning models, Jupyter Notebook for writing, debugging, and running deep learning code, Matplotlib, Seaborn, and Tensor Board are used for visualizing training metrics, model architectures, and data distributions. We used transfer learning to retrain a previously trained model for a new problem. Transfer learning offers numerous advantages, including the ability to finetune parameters quickly and easily to learn new tasks without defining a new network. 

Results Section: From the state-of-the-art models, DenseNet121 achieved a training accuracy of 99.73% validation accuracy of 99.16%, and testing of 98.33%% and MobileNetV2 scored a training accuracy of 99.64%, validation accuracy of 98.33%and testing of 96. 09%, While EfficientNetB7 scored a training accuracy of 99.82%, validation accuracy of 99.16%, and testing of 97.92%.

Discussion Section: The experimental results showed that DenseNet121 performed better than MobileNetV2 and EfficientNetB7 for field pea leaf disease detection. The experimental results also validated the use of transfer learning for field pea leaf disease classification.

---

## [Editor Report · Decision Letter 1]

11 Jul 2024

Field Pea Leaf Disease Classification Using a Deep Learning Approach

PONE-D-24-10799R1

Dear Dr. Girmaw,

We’re pleased to inform you that your manuscript has been judged scientifically suitable for publication and will be formally accepted for publication once it meets all outstanding technical requirements.

Kind regards,

Valentine Otang Ntui, Ph.D

Academic Editor

PLOS ONE
---

## [Editor Report · Acceptance letter]

16 Jul 2024

PONE-D-24-10799R1 

PLOS ONE

Dear Dr. Girmaw, 

I'm pleased to inform you that your manuscript has been deemed suitable for publication in PLOS ONE. Congratulations! Your manuscript is now being handed over to our production team.

Kind regards, 

on behalf of

Dr. Valentine Otang Ntui 

Academic Editor

PLOS ONE